# A Systematic Review of Internet of Things in Clinical Laboratories: Opportunities, Advantages, and Challenges

**DOI:** 10.3390/s22208051

**Published:** 2022-10-21

**Authors:** Tahir Munir, Muhammad Soomair Akbar, Sadia Ahmed, Azza Sarfraz, Zouina Sarfraz, Muzna Sarfraz, Miguel Felix, Ivan Cherrez-Ojeda

**Affiliations:** 1Department of Research, Nishtar Medical University, Multan 66000, Pakistan; 2Department of Research, Independent Medical College, Faisalabad 44000, Pakistan; 3Department of Research, Punjab Medical College, Faisalabad 38000, Pakistan; 4Department of Pediatrics and Child Health, The Aga Khan University, Karachi 74800, Pakistan; 5Department of Research and Publications, Fatima Jinnah Medical University, Lahore 54000, Pakistan; 6Department of Research, King Edward Medical University, Lahore 54000, Pakistan; 7Department of Pulmonology, Universidad Espíritu Santo, Samborondón 092301, Ecuador

**Keywords:** internet of things, cloud metrics, approaches, architecture, clinical laboratories, computing, blockchain, healthcare

## Abstract

The Internet of Things (IoT) is the network of physical objects embedded with sensors, software, electronics, and online connectivity systems. This study explores the role of IoT in clinical laboratory processes; this systematic review was conducted adhering to the PRISMA Statement 2020 guidelines. We included IoT models and applications across preanalytical, analytical, and postanalytical laboratory processes. PubMed, Cochrane Central, CINAHL Plus, Scopus, IEEE, and A.C.M. Digital library were searched between August 2015 to August 2022; the data were tabulated. Cohen’s coefficient of agreement was calculated to quantify inter-reviewer agreements; a total of 18 studies were included with Cohen’s coefficient computed to be 0.91. The included studies were divided into three classifications based on availability, including preanalytical, analytical, and postanalytical. The majority (77.8%) of the studies were real-tested. Communication-based approaches were the most common (83.3%), followed by application-based approaches (44.4%) and sensor-based approaches (33.3%) among the included studies. Open issues and challenges across the included studies included scalability, costs and energy consumption, interoperability, privacy and security, and performance issues. In this study, we identified, classified, and evaluated IoT applicability in clinical laboratory systems. This study presents pertinent findings for IoT development across clinical laboratory systems, for which it is essential that more rigorous and efficient testing and studies be conducted in the future.

## 1. Background

With a notable rise in technological advances, there are new possibilities to simplify our daily lives and improve the efficiency of services or production processes [1]. Digitalization and technological developments have been the epicenter of the fourth industrial revolution [2]. In recent years, the Internet of Things (IoT) as a network of connected devices is emerging as one of the key pillars of the ongoing technological revolution [3,4]. The IoT has gained prominence across different industries, and the combination of sensor data and analytical algorithms has enabled increased productivity, streamlined approaches, and leading products [5]. Popular technologies have been leveraged in IoT, including low-power wide-area networks (LPWAN) [6], cellular (3G/4G/5G) networks [7], Zigbee [8], Bluetooth low-energy (BLE) [9], WiFi [10], and radio frequency identification (RFID) [11] for the transfer of data to the cloud; the obtained data may be analyzed for meaningful, quick, and efficient decision making.

IoT is emerging as one of the most important technologies, and it has gained ground globally, being integrated steadily with applications in every industry, including healthcare. Already, IoT devices have been utilized in healthcare systems for continuous blood glucose monitoring, diagnosing apnea, and transmitting vital laboratory data from home to the hospital, documenting heart rate, skin temperature, movement monitoring, monitoring general health conditions, nutritional status, and analyzing elderly or infected patients [12,13,14,15,16]. There is uncertainty regarding the IoT and its utility and impact since it is in the early stages of implementation [17]. Many efforts have been made to categorize and advance IoT use in the healthcare sector [18,19,20,21,22]. However, certain application areas of IoT technologies, including clinical laboratories, are unclear on how to approach them; this is a pertinent indication that more research is required in this challenging field for important potential benefits for society [21]. In this regard, the implementation of IoT in clinical laboratories ought to be classified. For example, the typical clinical laboratory workflow begins with identifying and procuring the sample, analyzing the sample, and reporting the data to necessary stakeholders, e.g., patients and hospitals. IoT technology in clinical laboratories creates an integrated environment connecting patients with medical devices and healthcare personnel.

### 1.1. Defining IoT

Internet of Things (IoT) networks connect physical objects with a variety of interconnected sensors and actuator devices through computing systems [23,24]. Although there is no standard definition for the IoT, it has been defined multiple times in the literature [25,26,27,28]. The key objective of the IoT is to capture and share data from identifiable endpoints through a multilayer stack of technologies within its architecture. We use the following definition for the IoT: “…the interconnection of machines and devices through the internet, enabling the creation of data that can yield analytical insights and support new operations” [29].

### 1.2. IoT Architecture

Although there is no standard categorization for IoT architecture [30], it has been previously defined as a classic four-layer architecture, including the (i) perception layer, (ii) networking layer, (iii) middleware layer, and (iv) application/business layer [31]. As illustrated in Figure 1, the perception layer evaluates and gathers data from the physical environment. The network layer transfers and processes the environment data through gateways and the core network. The middleware layer stores data gathered from the network layer in a database. The application layer offers services from the perceived environment data, e.g., smart labs. The different layers are explained as follows [20,32,33]:*Perception layer:* The lowest layer of a conventional IoT architecture, this layer is also known as the *sensor*, *physical*, or *hardware* layer. It has sensors that collect physical signals or stimuli as data from various sensors, medical devices, or direct data entry from lab scientists. Once collected, the data are sent to the network layer.*Network layer:* This layer connects all smart devices and transmits clinical laboratory data from different users to the base station. Data may be securely transferred from other physical signals such as WiFi, Bluetooth, and Zigbee.*Middleware layer:* This layer, also known as the *processing layer*, stores, analyzes, and processes a huge set of data received from the network layer. Cloud computing is common for many technologies, whereas other databases may also be set up via this layer. Data analysis tools such as deep learning and lab protocol automation may be performed through this layer.*Application/business layer:* This layer delivers specific services to users, such as smart labs. It also includes applications, business models, and users’ privacy. This layer can be used for decision making by patients, hospitals, or lab scientists.

### 1.3. Approaches and Scopes of IoT

A robust taxonomy was proposed by Kashani et al. [20] for IoT-based healthcare systems, which we have adapted for this study and can be found in more detail [20]. Based on the taxonomy, five umbrella terms may be considered, including sensor-based approaches, resource-based approaches, communication-based approaches, application-based approaches, and security-based perspectives, explained as follows:*Sensor-based approaches*: this approach can be divided into two different categories, including wearable or environmental sensors.*Resource-based approaches*: this approach can be divided into five different categories, including scheduling, resource allocation, offloading, load balancing, and provisioning.*Communication-based approaches*: this approach can be divided into two categories, including technological and algorithmic.*Application-based systems*: this approach can be divided into two components: monitoring and recommender systems.*Security-based systems*: this approach can be divided into four components, including privacy, access control, confidentiality, and trust.

### 1.4. IoT Metrics

Nine metrics used to evaluate the IoT approaches for clinical laboratories in this paper are briefly defined as follows [20,34]:*Accuracy*: In clinical laboratory systems, accuracy is essential for all users, including patients and lab scientists. Data accuracy in clinical laboratories refers to low error margins and high precision levels. Such precision must be considered when analyzing a data set, protocol criteria, and lab technique [35].*Cost*: This metric refers to the total expenses the clinical laboratory service user requires. The prices may include communication, sensors, network, hardware, software, and data processing [34].*Energy*: This metric assesses the energy resources required for data signaling and processing. As the “things” are interconnected in IoT, communication is the most energy-consuming task, increasing costs and emitting carbon into the environment [34].*Interoperability*: Interoperability is “the ability of two or more systems or components to exchange data and use information” across clinical laboratories and hospitals [34]. Communication between different systems requires unified and wide-spectrum protocols for interoperability as many heterogeneous systems exist, e.g., sensors, hardware, software, and service.*Performance*: Issues that may arise are related to network quality of service (QoS) parameters associated with the network layer, including latency, delivery rate, and bandwidth usage. Other considerations defined as performance are load balancing, resource utilization, overhead, and computational time. This metric is necessary during all stages, including data collection, processing, and service delivery [36].*Scalability*: This metric refers to the ability of the clinical laboratories to increase the total capacity when there is an increase in service demand without reducing the performance. Clinical laboratory systems can improve scalability by increasing the capability of existing and additional hardware or services [37]. Different types of scalability include vertical, which refers to increasing the capacity of existing hardware or software by adding more resources, and horizontal, which refers to scaling out by adding more nodes to a system by connecting multiple hardware or software [37].*Security and privacy*: Security and privacy are essential as a building block of IoT to protect clinical lab data. The lack of standardization and regulation the IoT has, however, may precipitate many security and privacy issues. Users such as lab technicians may not have the necessary insight to control, service, update, and address concerns such as cyber-attacks which are required to preserve the integrity of the devices or services. Measures including architectural resilience to attacks, data authentication, access control, and user privacy must be established to preserve security and privacy [38,39].*Reliability*: The ability to successfully deliver services required in clinical laboratory systems is defined as reliability. This metric assesses the ability of clinical laboratory systems to perform relevant tasks in the given time and conditions [40].*Time*: As the name suggests, this metric assesses the time required for service delivery. Different measures include run-time, computational time, average response time, and latency in clinical laboratory systems [41].

### 1.5. Review of Related Literature

The related surveys, case studies, and systematic literature reviews conducted for IoT in healthcare are discussed to disclose the lack of comprehensive review and demonstrate the lack of robust studies approaching IoT in clinical laboratories taxonomically and systematically. The studies are summarized in Table 1.

#### 1.5.1. Surveys

Thilakarathne et al. [42] surveyed different uses of IoT healthcare technologies. The paper also discussed cloud computing, fog computing, blockchain, and big data that may be leveraged in healthcare systems. Furthermore, the authors examined the benefits of IoT-based healthcare services and applications. These benefits include real-time monitoring and reporting of data collected from patients, rapid analysis of data, high interoperability, reduced costs, quicker patient care, and quicker diagnosis. The challenges that the authors identified were underdeveloped technology systems in healthcare industries, lack of available resources, difficulty in regulating updates, lack of clear regulations, high energy consumption, integration across healthcare, difficulty in data analysis, high investment costs, lack of adequate training, and security concerns. However, the paper did not have a well-defined search strategy, and no taxonomy was provided.

Dang et al. [43] surveyed to analyze recent IoT components, applications, and market trends of IoT in healthcare. Three key components of IoT in healthcare were identified: topology or application scenarios, structure or physical components, and platform or frameworks. Networks commonly used in healthcare were short-range and medium-range communications, and cloud computing was explored. Differences in cloud and fog computing were listed, and fog-assisted systems were suggested to be a stronger candidate for healthcare IoT ecosystems. The authors discussed open challenges of the three key components, including topology, structure, and platform. However, the paper did not have a well-defined search strategy, and no taxonomy was given.

Islam et al. [44] presented a survey of the IoT in healthcare, including applications and smartphone applications for general use. The authors also discussed security requirements and challenges followed by a threat model. Based on the threat model, an attack taxonomy was suggested, including information-, host-, and network-specific compromise. However, the search strategy was not defined by the authors.

Dhanvijay et al. [45] surveyed the security and privacy features of IoT-based healthcare solutions. Various parameters of security and privacy were reviewed. An open challenge was the lack of well-defined system architecture. The authors did not define a search strategy or present a taxonomy.

#### 1.5.2. Case Studies

Yuehong et al. [46] presented a case study prototype of an IoT-based smart rehabilitation system. The average similarity between the prescription given by the doctor and the system was 87.9%, and the cost was 53.1% lower. The paper also summarized the application history of IoT technology in healthcare industries, identification technology, communication and location technology, sensing technology, and service-oriented architecture. Afterward, the implementation methodologies, such as resource, knowledge, and big data management, were reviewed. Additionally, strategies for telehealth and telerehabilitation systems were discussed. However, the paper’s selection process was unclear, and no taxonomy was provided.

Habibzadeh et al. presented a case study based on their survey of existing and emerging technologies in healthcare. In this case, the patient has Parkinson’s and Huntington’s, and multiple sensing modalities as a single sensor were used. An IoT architectural framework was developed based on the patient’s specific needs. Open challenges, including system security and data privacy, were considered the highest priorities by the authors. There was no taxonomy provided.

#### 1.5.3. Systematic Literature Reviews

Kashani et al. [20] conducted a notable systematic review of healthcare and IoT. The study presented a comprehensive taxonomy and divided the technologies into five different categories presented in Section 1.3. The authors reviewed the evaluation techniques, tools, and metrics. Open issues were considered, including interoperability, real testbed implementation, scalability, and mobility.

Dwivedi et al. [13] systematically reviewed IoT applications in healthcare. Different technologies integrated with IoT are virtual reality, mixed reality, augmented reality, parallel fog, edge, cloud computing, 5G networking, big data visualization and analytics (BDVA), artificial intelligence, and blockchain. Smart healthcare consists of smart hospitals, smart medication, digital biomarkers, smart operating rooms, aerial thermal scanners, 3D printing, virtual planning of surgeries, smart applications, e-learning, teledentistry, virtual assistants, ambient assisted living, and adverse drug reaction detection. Open challenges discussed were privacy and security, data management, scalability, interoperability, cost-efficacy, power consumption, and environmental impact.

Bolhasani et al. [48] systematically presented taxonomies, categories, and environments for the deep learning application of IoT in healthcare. Four major taxonomies of deep learning were identified, including medical diagnosis and differentiation, home-based and personal healthcare, disease prediction, and human behavior recognition.

Forum et al. [49] systematically explored blockchain technology and demonstrated two case studies to examine the role of blockchain in health care. Blockchain technology could be used in clinical trial management, such as smart-contract applications, participant-controlled data access, trustless protocols, and data validity. Other applications were electronic health records (E.H.R.s), patient-centered interoperability, remote patient monitoring, and management of clinical trial data. While the authors did not discuss open issues with blockchain, cybersecurity as a current issue in healthcare management systems was addressed with the ongoing digitalization.

Sadoughi et al. [50] mapped the current IoT developments in medicine to map out the recent experimental and practical use of IoT in sub-fields. The fields receiving the most IoT attention were neurology, cardiology, and psychiatry/psychology. The authors did not address any open issues in specific areas of medicine.

Nasiri et al. [51] presented the security requirements of IoT in healthcare systems in a systematic review format; they reviewed four databases and identified key features and concepts associated with the security requirements in healthcare systems. Two categories of security requirements were identified, including cyber security and cyber resiliency, both considered essential to establish trustworthiness in the IoT-based healthcare system.

### 1.6. Aims and Objectives

A comprehensive systematic review of IoT in clinical laboratories has not been conducted. With the implementation of IoT in healthcare already in motion, a robust understanding of the clinical laboratories’ IoT adaptation is necessary. We believe that a comprehensive review inclusive of IoT priority areas in laboratories will provide an integrated perspective and serve as a repository for knowledge. In the present study, we aimed to collate all existing evidence, classifications, and applications of IoT in different clinical laboratory processes. Our objectives were to *identify*, *classify*, and *evaluate* the current IoT literature to inform our understanding of the IoT-supported clinical laboratory adoption and implementation. 

### 1.7. Motivation for This Study

The motivation to conduct an S.L.R. on IoT in clinical laboratories is that until now, there has been no systematic review of IoT-based clinical laboratories. Most existing papers focus on healthcare systems holistically, and these studies do not analyze trends, open issues, and challenges of IoT in clinical laboratories. Many of the studies summarized in Section 1.4 do not provide open challenges. Ours is the first paper to conduct an S.L.R. to explore IoT in clinical laboratories. Therefore, we conducted a comprehensive review to answer the research questions and associated motivations in the present study (Table 2). As highlighted in the assessment of the current literature, existing surveys, case studies, and systematic literature reviews that are conducted for IoT in healthcare do not depict robustness in approaching IoT in the clinical laboratories’ taxonomy. Hence, firstly, this study aims to robustly identify the different IoT categories and metrics to integrate them into clinical laboratory systems. Secondly, this study also wishes to identify and reveal the tools and frameworks that can advance the current status of IoT in clinical laboratory systems. Thirdly, this study will also reveal the opportunities present in this field to have a coherent understanding of IoT applications. Lastly, this paper will identify the research considerations required to apply IoT in clinical laborites.

### 1.8. Structure

This structured, systematic review aims to identify, classify, and evaluate IoT applications in clinical laboratory systems. Section 1 describes the introduction, the definition of IoT in Section 1.1, IoT architecture in Section 1.2, approaches and scopes of IoT in Section 1.3, IoT metrics in Section 1.4, a review of the related literature in Section 1.5, the aim and objective in Section 1.6, and motivation for this study in Section 1.7. Section 2 details the methods, including the theoretical framework and search strategy in Section 2.1, eligibility criteria in Section 2.2, data extraction and synthesis in Section 2.3, and critical appraisal and bias assessment in Section 2.4. Section 3 describes the results, including the risk of bias in Section 3.1, IoT in the preclinical laboratory phase in Section 3.2, IoT in the clinical laboratory phase in Section 3.3, and IoT in the postanalytical laboratory phase in Section 3.4. Section 4 details the discussion, including open concerns, challenges, and future trends in Section 4.1. The conclusion is summarized in Section 5.

## 2. Materials and Methods

### 2.1. Theoretical Framework and Search Strategy

Webster and Watson (2002) described a concept-centric approach to developing ad hoc classification systems in which categories are used to describe areas of literature [52]. We used a concept-centric approach, focusing on IoT in laboratories, and we explored the literature related to these data to support the development of the theoretical framework and conceptual model. We used a similar step-by-step approach to Leidner and Kayworth (2006) to develop a framework for our search strategy [53]. A five-step approach was developed to identify the literature **(**Figure 2). The study was conducted in adherence to the preferred reporting items for systematic reviews and meta-analyses (PRISMA) statement 2020 guidelines [54].

First, we used the key terms *Internet of Things*, *IoT*, and *laboratory* to search across databases, including Pubmed, Scopus, and Google Scholar, conducted by three reviewers (T.M., S.U., and M.S.U.). Three reviewers initially screened the key terms in the title and abstract in the first review stage (T.M., S.U., and M.S.U.). The full keyword strings are attached in Appendix A. In the second stage, full texts were assessed for eligibility against our inclusion criteria with consensus from five reviewers (T.M, S.U., M.S.U., AS, and Z.S.). Searches were refined iteratively as there was a paucity of empirical studies. This means a different combination of key terms was run to find the most relevant literature for inclusion. Second, a search of leading scholarly journals in information systems (I.S.) and technology was conducted at the Institute of Electrical and Electronics Engineers (IEEE) Xplore and A.C.M. Digital Library. Similar selection processes were applied in two stages as that for databases. In the first stage, the titles and abstracts were screened for potential eligibility by three reviewers (T.M., S.U., and M.S.U). The second stage was screening full texts for inclusion in our study, which five reviewers assessed (T.M., S.U., M.S.U., AS, and Z.S.), and a consensus was reached. Papers from reputable sources were selected for our paper.

Third, references to key papers identified in the first stage of screening that were within the scope of our topic were further explored to find any relevant literature for inclusion (Webster and Watson 2002). We screened these papers in the bibliographies in two stages, similar to our first two steps. Fourth, articles related to IoT priority areas in laboratory settings were reviewed analytically, and three functions of IoT were found as focal points—preanalytical, analytical, and postanalytical. These functions were considered priority areas for IoT in laboratory settings. Furthermore, the identified literature was analyzed critically to provide IoT opportunities, advantages, and challenges once studies after studies were assessed against the inclusion criteria. Fifth, for identified focal points of identified papers in the first stage of screening, we redefined them after the final stage of screening.

Given that IoT is an emerging technology, research-based literature is somewhat limited. Identified articles were included if they were relevant to the theme of the paper. Laboratory processes were grouped as preanalytical, analytical, and postanalytical phases, and further description of specific functions was tabulated during the second screening stage. Studies were not considered in both screening stages if they were irrelevant to the theme of the paper.

### 2.2. Eligibility Criteria

*Inclusion Criteria:* Research articles (trials, observational studies, reports, and concept articles) that covered the application of IoT in clinical laboratory processes were included. Research articles from 2015 to 2022 were included in well-reputable journals.

*Exclusion Criteria:* Papers that were not concept papers or primary research articles and did not belong to any of the focal points of IoT in laboratory processes were excluded. Books, conference papers, symposiums, commentaries, and review articles were excluded. Non-English studies were omitted as well. Their bibliographies were screened to identify any relevant papers pertaining to our theme.

### 2.3. Data Extraction and Synthesis

We stored all studies identified from these databases in Endnote X9 (Clarivate Analytics, London, UK). The duplicates were removed using the Endnote X9 deduplication tool. There were no language restrictions; non-English studies were not excluded. The titles and abstracts of all shortlisted studies from the databases and reference lists of these studies were independently screened by all researchers; in the case of disagreements, the final reviewer was present to resolve discrepancies. All researchers extracted the data onto a shared, customized spreadsheet under the following categories: primary author, year, country, journal, framework/model, user(s), tool(s) and evaluation environment(s), main idea, evaluation technique(s), performance analysis, opportunity, advantage(s), and challenge(s).

### 2.4. Critical Appraisal and Bias Assessment

Cohen’s coefficient of agreement was calculated to quantify the inter-reviewer agreements. Joanna Briggs Institute’s (J.B.I.) critical appraisal tool was utilized to critique the research evidence [55]. The methodological quality of studies was assessed to determine the extent to which the study addressed the possibility of bias in the design, conduct, and analysis. All papers selected for inclusion in this systematic review were subjected to rigorous appraisal by two critical appraisers (A.S. and Z.S.). The results of this appraisal will help inform the interpretation and synthesis of the study results.

## 3. Results

In phase 1, 2384 studies and records were identified in the original and umbrella search (2298 studies were identified from databases, 73 from websites, and 13 from citation searching). Of these studies and records, 263 were duplicates. In phase 2, a total of 2121 studies were screened only for titles and abstracts (2035 studies were screened from databases; 13 records were assessed for eligibility from websites and citation searching); in this phase, 2080 were excluded (post title, abstract, and record screening) as they did not fit with the objectives of this study/inclusion criteria (i.e., study types and content). The studies were stored in Endnote X9 for perusal by the first two authors, with the third author present for any disagreements. In phase 3, 41 studies were sought for full-text review, of which 28 were identified from databases, and 13 records were identified via websites and citation searching. Of the full-text studies retrieved, 23 studies were omitted as they met the exclusion criteria as appended earlier (Appendix A). Finally, 18 studies were included in this systematic review adhering to the aims/objectives of this study and met the inclusion criteria. Cohen’s coefficient of the inter-reviewer agreement was computed to be 0.91. The PRISMA flowchart is attached in Figure 3.

### 3.1. Risk of Bias

The study-by-study critical appraisal based on the J.B.I. tool is attached to Appendix A. All 18 studies had the source of the text identified. Concerning the source of the text having standing in the field of expertise, 14 had standing, 3 were unclear, and 1 did not have standing in the field of expertise. On assessing the interests of the relevant population as the central focus of the text, 14 were relevant, 2 were unclear, and 2 were irrelevant. When assessing if the stated position resulted from analytical processes and if logic was present in the text, 15 studies checked yes, 2 were unclear, and 1 was not. Fourteen studies referenced the extant literature, two were unclear, and two did not reference the extant literature. Incongruence with the literature/sources being logically defended was seen in 15 studies, 2 were unclear, and 1 was incongruent in this section (Appendix A).

### 3.2. IoT in the Preanalytic Laboratory Phase

Reviewing the selected preanalytical laboratory phase articles indicates that there are clinical laboratory supports consisting of temperature and humidity sensors for laboratory monitoring and electromagnetic sensors for liquid testing, including Kang et al. [56] and Galindo-Romera et al. [57]. One study focused on the transportation of potential laboratory specimens supported by vehicle user sensors by Alam et al. [58]. The studies classified as preanalytic are discussed (Table 3).

#### 3.2.1. Clinical Laboratory Monitoring and Support

Kang et al. [56] presented an IoT-equipped monitoring model for an on-premise hospital’s laboratory, focusing on temperature and humidity sensors for refrigerators and freezers for quality control. The IoT model consisted of four components: sensor management, temperature, and humidity monitoring, alerting and reporting, and indoor mapping. When the temperature fell outside the range of the preset tolerance limit, the Bluetooth-enabled low-energy (BLE) communication platform generated an alert on a dedicated web application.

Galindo-Romera et al. [57] proposed a low-cost, portable IoT Reader (IoT-R) for passive wireless electromagnetic sensors that obtain the relative dielectric permittivity of different liquids under test (LUT). Two elements are present: a passive wireless electromagnetic sensor that senses different magnitudes and an IoT-R, which generates the radiofrequency signal to send the measurement through the Internet. Such a technology may be leveraged to detect the substance that has been leaked in the laboratory, e.g., harmful substances.

#### 3.2.2. Smart Vehicle and Transport

Alam et al. [58] proposed the social internet of vehicles (SUV) that establishes communication within vehicles by forming vehicular ad hoc networks (VANETs). The cyber-physical cloud computing (CPCC) system formed the IoT interface, including vehicles, roadside infrastructures, houses, and the cloud. The SUV consists of dynamic nodes, which are on-board diagnostics (OBD) present in vehicles; static nodes, which are home-based units (HBU) to which all vehicles are connected; and roadside units (RSU) representing the roadside infrastructure. Each OBD is connected to one or more wireless adaptors for communication with surrounding OBDs and RSUs. The OBD has smartphone-supported sensor devices that obtain the user’s status, such as physical properties, mental state, emotional state, abilities, and characteristics.

### 3.3. IoT in the Analytic Laboratory Phase

Six studies focused on point-of-care (POC) diagnostic testing enabled with cloud-assisted IoT, including Kalasin et al. [59], Kalasin et al. [60], Alonso et al. [61], Ma et al. [62], Zhu et al. [63], and Wang et al. [64]. Bibi et al. [65] applied a deep learning algorithm with IoT to diagnose leukemia and subtypes with blood smears. Two studies support high-throughput analysis and purification in laboratories through IoT-supported models, including Neil et al. [66] and Shumate et al. [67]. Two studies demonstrate virtual assistance and central automation, including Austerjost et al. [68] and Porr et al. [69]. The studies incorporating IoT within analytic laboratory processes are discussed (Table 3).

#### 3.3.1. IoT-Supported Point-of-Care Testing

Kalasin et al. [1,59] and Kalasin et al. [2,60] both demonstrated the chemical transduction and Bluetooth-enabled point-of-care (POC) technology: the former for the direct determination of salivary creatinine to monitor kidney disease, whereas the latter for the concurrent monitoring of heat-stress sweat creatinine. The IoT model of both studies comprises sensors/sensing platforms and an electrochemical base. The evaluation environment and tools consist of (i) a smartphone, (ii) PSRAM-WiFi, (iii) Bluetooth, and (iv) analytical-grade chemicals. The core applications of the model and tools are seen in the chemical transduction of bodily fluids (i.e., saliva, body fluid). Both studies had good sensitivity and efficacy (97.2% and 96.3%, respectively). Both POC technologies can be used in underequipped communities to test for kidney or other disorders.

Alonso et al. [61] presented an IoT-based malaria POC testing. The framework was based on a field-programmable gate array integrated with IoT. The model consisted of a field-programmable gate array (FPGA) with IoT capabilities. The tools comprise a custom single-photon avalanche diodes (SPADs) camera, IoT-enabled remote control, and the ELISA system. The sensitivity was measured with the photon detection probability of the SPADs camera always below 15%. While the PoC malaria testing kit is cost-effective and convenient, scalability costs and variable sensitivity are two important challenges [61].

Ma et al. [62] used smartphones for IoT-based real-time diagnosis of sleep apnea. The IoT model was based on a support vector machine (SVM) to remotely diagnose obstructive sleep apnea syndrome (OSAS) with the help of data, specifically blood oxygen saturation (SpO2). Overall, the sensitivity and specificity of their system were 87.6% and 94.1%, respectively.

Zhu et al. [63] conducted PoC testing for the dengue virus. The authors’ framework enabled PCR chips within the IoT system, which aided in the rapid diagnosis of the dengue virus through DNA or RNA detection of the viral strand. In total, 40 cycles to detect the DNA of the viral strand required 34 min. While IoT-enabled POC testing for the dengue virus has excellent potential to allow live reporting via a smartphone platform, there were concerns for sensitivity variability depending on the local systems used and misuse by the health workers [63].

Wang et al. [64] used cloud-assisted IoT for PoC testing of lupus nephritis. The tools consist of a strip test, an IoT device, and a cloud server. The sensitivity and specificity were both higher than 80%. However, due to the system’s complexity, misuse by healthcare personnel may occur, along with variable sensitivity and variable reliability [64].

Bibi et al. [65] applied IoT-based systems to detect and classify leukemia. Their model consisted of IoT-enabled microscopes, which were then used for blood-smeared images; the datasets were moved to the leukemia cloud, and augmented with the dense convolutional neural network (DenseNet-121) and residual convolutional neural network (ResNet-34) models. The simulation was excellent compared to existing machine-learning algorithms for leukemia subtype classification.

#### 3.3.2. Virtual Assistance and Quality Control of Central Laboratory Services

Neil et al. [66] proposed end-to-end clinical laboratory sample tracking in a high-throughput analysis and purification laboratory through a custom IoT-supported model consisting of LED strip-supported audiovisual feedback and linear and 2D barcoded containers. The BlinkStick LED strip is attached to the network and is automated via its continuous operation mode, serving as the primary visual feedback device.

Shumate et al. [67] applied a quality control platform termed just another monitoring system (JAMS) dispenser. The model consists of an automated, accurate dispenser enabled via IoT to provide gravimetric summaries of liquid dispensing and obtain data for future analysis. The system leveraged a sensor to gravimetrically get the amount of liquid dispersed which does not affect the HTS. The model was shown to increase the overall laboratory function and maintain quality by accurately evaluating the dispense rate of liquids.

Austerjost et al. [68] proposed a voice user interface (VUI) to control laboratory devices and read specific data. These laboratory-based devices apply a voice user interface to control laboratory instruments without the need for the sense of touch. The possible applications of VUI include a stepwise reading of standard operating procedures (SOPs) and recipes, reciting chemical substance or reaction parameters in control, and reading out of laboratory devices and sensors.

Porr et al. [69] implemented a micro-service-based automated digitalized central laboratory server using the open-source Standardization in Lab Automation 2 standard (SILA2) communication standard. The central lab server channel offers process management of principle steps, including database records, results, and protocols.

### 3.4. IoT in the Postanalytical Laboratory Phase

One study by Guo et al. [70] on disseminating a lateral flow immunoassay (LFIA) test status for an infectious disease outbreak and leveraged cloud-based patient data storage is discussed. Three studies focusing on blockchain technology to create smart healthcare delivery models, including Celesti et al. [71], El Majdoubi et al. [72], and Ochôa et al. [73], are discussed (Table 3).

#### 3.4.1. 5G-Enabled Data Sharing

Guo et al. [70] proposed a medical device and software application for 5G-enabled fluorescence sensors detecting COVID-19. The sensor can detect the spike (S) protein and nucleocapsid (N) protein of SARS-CoV-2. The device uses mesoporous silica encapsulated up-conversion nanoparticles (UCNPs@mSiO_2_) labeled lateral flow immunoassay (LFIA). In clinical use, the sensor tested viral culture as clinical samples. The sensor is IoT-enabled, which means it is accessible by hardware devices such as 5G smartphones and personal computers. Using the fog layer of the network and the 5G cloud server, the patients and their families will obtain medical data, reducing the burden of visiting hospitals for testing and receiving results. The device is embedded with fuzzy logic and deep learning algorithms into the system to intelligently account for event-driven emergencies among patients.

#### 3.4.2. Blockchain-Enabled Data Sharing

Celesti et al. (2020) [71] proposed telemedical laboratory services where IoT-enabled medical devices were synced to hospital clouds for consultation and validation. Blood tests conducted in a laboratory (e.g., complete blood count, blood glucose) performed by technicians can be directly reported to hospitals through IoT devices interconnected via a hospital cloud system to doctors for consultation. Key personnel involved were nurses, technicians, medical doctors, federal hospitals, and public and hybrid blockchain using the Ethereum platform in the proposed model.

El Majdoubi et al. (2021) [72] proposed a smart healthcare solution called SmartMedChain that used an end-to-end blockchain framework for data sharing. The system builds on the existing blockchain framework built on Hyperledger Fabric by creating an interplanetary file system (IFPS) that stores data. When integrated with electronic health records (EHRs) and medical IoT data, the model leverages IoT and cloud-based services to keep up with high volumes of patient data (diagnosis, medical laboratory reports, and insurance documents).

Ochôa et al. [73] devised a smart middleware contract using the Ethereum blockchain, ensuring user privacy. The model, UbiPri, facilitates three layers, including the blockchain, communication, and application layer, and caters to individual privacy settings through user, environment, and smart device contracts.

## 4. Discussion

IoT may help facilitate various laboratory and clinical processes, as explored in this study. Applications of IoT in clinical laboratories were elaborated on in the preceding section. We analyzed the result of the SLR based on the methods defined in Section 2. Figure 4 depicts the different journals where 18 of the articles until 2022 are published. The most frequent journals were *Sensors* (Basel) (n = 4), *SLAS technology* (n = 3), *ACS Biomaterials Science and Engineering* (n = 2), and *Biosensors and Bioinformatics* (n = 2), where more than one of the included primary articles were published. Figure 5 demonstrates the year of publication of all the included investigations.

Based on the research questions in Section 1.5, we respond to RQ1, RQ2, and RQ3 as follows:


**RQ1: What are the different categories for IoT integration in clinical laboratory systems?**


Figure 6 illustrates the different categories and applications for IoT in clinical laboratories based on the suggested taxonomy in Section 2.1. Three application fields were considered for IoT integration in clinical laboratory systems: preanalytical, analytic, and postanalytical.

In the preanalytical phase, the special functions are focused on primarily temperature monitoring of vehicle-based systems, which serve as intelligent transport systems. These are home- or hospital-based sample collection tools where samples can be transferred to the laboratory. Secondarily, this phase also focuses on the wireless electromagnetic sensors that are embedded within laboratory instruments to avoid direct contact with samples during the collection phase. Thirdly, sensors for temperature and humidity are placed in freezers and refrigerators to prevent inaccuracies in data due to the pre-handling of the samples.

In the analytic phase, there are four areas of functional concern. Firstly, IoT-enabled point-of-care tests are conducted in the field. Secondly, IoT can enable satellite communication to detect medical emergencies in real time. Thirdly, IoT can diagnose leukemia and sleep apnea by monitoring real-time physiological and remote data. Lastly, IoT can enable voice user interface and 2D barcodes for standard operating procedures, specific sensors and data in laboratories, and for controlled reaction parameters.

In the postanalytical phase, IoT allows the sharing of data using cloud servers between hospitals, laboratories, and other beneficiaries. Moreover, IoT-enabled reporting permits laboratory results to be shared with patients in a real-time syncing manner.


**RQ2: What are the existing tools, evaluation techniques, and approaches that enable IoT in clinical laboratory systems?**


The common tools and environments used were Python, Java, and cellular networks. A total of 78% of the included papers were tested in real testbeds, whereas only 22% were not. Figure 7 illustrates the percentage of evaluation techniques in all investigations. Of the included papers, the approaches and scopes used were assessed as per those enlisted in Section 1.3, summarized in Table 3. The most frequent approach was communication-based, whereby 83.3% of the included investigations incorporated this. Application-based approaches were seen in 44.4% of the included investigations. The next most frequent was sensor-based approaches which were present in 33.3% of the included investigations. Both resource-based and security-based approaches were considered in 27.7% and 16.6% of the included investigations. A total of 77.7% of the included studies employed more than one approach. Figure 7 illustrates the percentages of the different approaches in the included studies.


**RQ3: What opportunities are available with the existing IoT techniques in clinical laboratory systems?**


Figure 8 illustrates four different phases that form a smart clinical laboratory extrapolated from the primary studies in our study.

In the *first* phase of sample barcoding, the collected and analyzed clinical samples may be tracked through barcodes. Test tracking through 2D or linear barcodes allows for tracking of the samples from end to end, which permits management to correct any deviations in real-time and sample safety in transit.

In the *second* phase or sample scheduling and tracking, the transportation of clinical blood samples forms a critical part of the preanalytical phase. In this phase, automated transport systems with sample integrity monitoring and temperature surveillance contribute to continuous quality control. Additionally, clinical demands of high turnaround rates of sample collection can be managed with smart transport. For instance, by placing temperature sensors within coolers, clinical laboratories may obtain real-time data about the location and sample quality.

In the *third* phase or sample processing and quality control, the central laboratory server can be automated through virtual assistance with a centralized lab server. The central lab server communicates and contains the database for every measurement, task, and result created or used within the lab. The interface acts as a central entry point for the management of the digital lab, which also serves as a guide or assistance for the lab workers, such as automated calculations and safety information. Lab scientists do not need to conduct calculations, data analysis, and documentation as the interface can carry out these processes. Different tasks may include data-recording automation, supply restocks automation, and predictive maintenance of machinery. In predictive maintenance, equipment such as freezers and fridges can be continuously monitored for quality and regulatory compliance with alert systems. Additionally, computing systems can gather information from laboratory equipment, such as instruments and equipment, which is then fine-tuned, monitored, and analyzed. This information can help identify potential failures before they can happen, such as fluctuations in laboratory environmental conditions, which can be used to trigger alarms through IoT-enabled temperature, humidity, pressure, and light sensors.

In the *fourth* phase, the results of point-of-care tests (PoCT) may be stored in the cloud. Detection of biochemicals within biofluid is one example of PoCT for continuous diagnosis throughout day-to-day activities in this phase. Another method of reporting is blockchain which may be leveraged to provide telemedical laboratory services for automated and integrated healthcare delivery for patients.


**RQ4: What are the key advantages, open challenges, concerns, and future trends in IoT in clinical laboratory systems?**


Based on Table 4, accuracy (44.4%), performance (44.4%), and time (38.8%) are the most common metrics that are considered strengths or *advantages* of the IoT in the included primary studies. The least common metrics considered advantages by authors of the papers are reliability (11.1%), scalability (16.6%), and energy (16.6%). The advantages and challenges of IoT technologies based on key metrics defined in Section 1.4 are visually depicted in Figure 9 and Figure 10 using scatter plots. When considering the advantages represented in the studies, these include accuracy (44.4%), time (44.4%), performance (44.4%), interoperability (33.3%), and security/privacy (33.3%) as the most common ones; scalability is represented at 16.7% (Figure 9). Scalability (55.5%), costs (33.3%), and interoperability (27.7%) are the most common metrics that are considered challenges or disadvantages of the IoT in the included primary studies. Among these, the least common metrics considered challenges in the studies are reliability (5.5%) and security and privacy (5.5%) (Figure 10).

### 4.1. Open Concerns, Challenges, and Future Trends

Significant problems in IoT-based laboratories must be considered and have not been addressed. The collected data we analyzed in this review suggest common challenges, including scalability, costs, and interoperability. In this section, to answer RQ3, open challenges, concerns, and future trends have been presented. A graphic illustration of open challenges/concerns and future trends is illustrated in Figure 11 and Figure 12, respectively.

#### 4.1.1. Open Challenges and Concerns

*Scalability*: Based on the reviewed literature (Galindo-Romera et al. [57], Alam et al. [58], Ma et al. [62], Bibi et al. [65], Shumate et al. [67], Austerjost et al. [55], Porr et al. [56], Guo et al. [57], Celesti et al. [58], and El Majdoubi et al. [59]), scalability is a major open challenge for IoT in clinical laboratories. In the reviewed studies, proposed approaches would be feasible in a small capacity. However, with the expansion of the system’s scale, the system’s ability to keep up with the rapid turnaround rates seems to be a challenge. A key open issue found in this study is the scalability and adoption of IoT systems for expanding IoT systems across and within clinical laboratories.

*Costs and energy consumption*: Based on the studied literature (Kang et al. [56], Galindo-Romera et al. [57], Shumate et al. [67], Celesti et al. [58], El Majdoubi et al. [59], Ochôa et al. [60]), energy consumption results in high operational costs and also results in massive carbon footprint. The technologies are powered by energy, and a growing number of sensors and devices will require higher consumption. Optimized solutions to reduce energy consumption will be required to reduce costs and save energy resources. A prominent open issue found in this study is resource management, which includes costs and energy consumption.

*Interoperability*: According to the included literature (Kang et al. [56], Kalasin et al. [46], Kalasin et al. [47], Bibi et al. [52], Neil et al. [53]), interoperability is another crucial component that is considered an open challenge for IoT implementation in clinical laboratories. Interoperability is related to the lack of standardization of IoT due to the lack of implementation of standard rules and regulations for compatible interfaces and protocols across devices. Due to the lack of consensus on communication protocol and standards, a new IoT device may not be compatible with the ecosystem. This lack of conformity may lead to reduced interoperability. Therefore, a significant open challenge found in this study is the interoperability of different IoT systems.

*Privacy and security:* In a paper from the reviewed literature, by Guo et al. [70], privacy and security were not discussed explicitly, and many technological innovations suggested by the authors were shown to be secure. Regardless, the high volume of data concerning patients’ clinical reports will require trust as the nature of this data is confidential. All stakeholders, e.g., hospitals, healthcare professionals, patients, and lab scientists, are not supposed to have unauthorized access to these data. With future adoption in clinical laboratory systems, it is important to consider and reduce the risk of cyber and physical attacks such as data hacking and increase data protection. Thus far, the privacy and trust of the users can be considered an open issue.

*Performance:* In the reviewed literature (Kalasin et al. [55], Kalasin et al. [56], Alonso et al. [57], Zhu et al. [59], and Ochôa et al. [73]), the performance of the IoT systems was considered. For the models to work optimally, different concerns can be present, such as cost and computational lag, whereas other factors, such as reliability, can be a tradeoff for these concerns. Various challenges in performance may be considered based on the objectives of the IoT systems and remain an open issue.

#### 4.1.2. Future Trends

*Networks:* A typical IoT ecosystem has a wide range of networks that range from short-range (e.g., WiFi, Zigbee, and Bluetooth) to long-range (e.g., cellular and personal wireless communication networks) [74]. The use of ultra-wideband short-range networks (e.g., radio technology) can help reduce power consumption when used with sensor devices apt for clinical laboratory systems [75]. Software-defined networks (SDNs) are a recent networking architecture approach that allows central programming of the entire network and its devices using open application programming interfaces (APIs) [76]. With SDNs, a virtual network can be created, or a traditional hardware device, e.g., routers, can be controlled; the SDN-defined network function virtualization (NFV) architecture offers optimized provision, deployment, and centralized management of the virtual network [77]. SDN-NFV architecture is an interesting area for future research.

*Cloud and Grid computing:* Cloud servers have been used in IoT ecosystems within healthcare to obtain real-time information, store and analyze large volumes of data, and distribute different levels of service based on users outside the cloud [78]. Cloud computing can provide consistent access to data in clinical laboratory systems from any connected device across the Internet [21]. Cloud computing has evolved from grid computing which functions similarly to a physical public utility such as electricity and public telephone networks [79]. Grid or cluster computing is being given attention for its implementation in healthcare as it may reduce processing times, provide access to healthcare and academic professionals, and have global applicability [80]. Grid computing, while not a new paradigm, can provide decentralized storage for different users to use free of cost, and it has interesting potential in clinical laboratory systems.

*Fog computing:* Fog computing, also known as edge computing, is proposed to enable computing at the edge of the network, e.g., commercial edge routers improve processor speed, the number of cores, and built-in network storage [81,82,83,84]. Fog is a layer of a distributed network closely related to cloud computing and is an extension of the cloud computing paradigm. A fog layer addresses the concerns that manifest with cloud computing through computation, storage, and networking services between end devices and Cloud servers [81,82]. With fog computing, high latency and low response time may be addressed. This novel paradigm is emerging and may be able to overcome the challenges of cloud computing.

*Big data visualization and analytics:* Huge volumes of data obtained in the cloud can be managed using enhanced data analytics capability. For instance, data from different hardware and software can be collected through big data tools and stored in the cloud. Big data analytics can integrate data in a semistructured, unstructured, and structured format and conduct analysis [85]. Currently, many sensors and devices are operating at a very limited computational capacity which can be addressed by big data analytics [86]. Big data analytics may address and provide intelligent integration for decision-making in the context of patient clinical data [86,87]. However, certain challenges need to be addressed, including the complexity of the data in healthcare systems [88], such as transmission, storage, quality of data, privacy, and governance [89,90]. In the future, the collaborative approach between cloud computing and big data analytics has the potential to address the challenges associated with large volumes of data, such as in clinical laboratory systems [91].

*Security considerations:* The current status of security and privacy in IoT ecosystems is that it remains underdeveloped. Separate from the basic layers of IoT architecture, the IoT ecosystem must have confidentiality, integrity, authentication, availability, resiliency, fault tolerance, and self-healing [92,93,94]. Yaacoub et al. proposed three new layers within the IoT architecture that may be implemented, including (i) the accuracy layer (trust sub-layer): using trusted third parties, building patient trust, and being less error-prone; (ii) the prevention layer (authentication sub-layer and privacy sub-layer): authenticating user and device, controlling access, authenticating source authentication, and ensuring privacy and anonymity; (iii) defensive layer (detection sub-layer, correction sub-layer): integrity of data systems, antimalware software, firewalls, backing data, alternative devices, and configuration. For instance, the prevention layer will troubleshoot a zero-day exploit in a medical device that can be used to injure or kill someone without detection [95]. Cyber vulnerabilities have been a major challenge as health data has liability and significant risks [96,97]. IoT cannot be safely adopted in clinical laboratory systems without concerned authorities addressing risks and placing security measures. Hence, security and privacy remain an open challenge and key merit of attention for future studies.

*Blockchain and Online Social Networks:* IoT ecosystems in healthcare typically have four layers of architecture, from sensors to Fog devices and a centralized cloud server. Privacy concerns arise since cloud servers involve third parties. Additionally, data flow issues may create a bottleneck that would require regular updates from a maintenance perspective. This has resulted in advances for IoT through blockchain for cloud and fog IoT in the context of healthcare, such as intelligent transport and eHealth [98]. Blockchain’s prominent features that may enhance clinical laboratory systems’ functionality include decentralization, security, privacy, transparency, and immutability [99]. Blockchain seems promising as it provides “smart contracts” that result in distributed anonymous communication without a centralized authority [100]. Blockchain-supported smart laboratories may preserve end-users’ anonymity while also ensuring transparency [101]. Blockchain does not support the deletion or alteration of information from blocks, making it an attractive technology for clinical laboratory systems [99]. This concept is similar to online social networks that connect clinicians, pharmacists, patients, and medical devices through dedicated software interfaces. While the paradigm is emerging, it has excellent potential to apply to clinical laboratory systems [102,103]. The issues with online social networks can be addressed through blockchain, such as end-user privacy and security [104]. We illustrate a smart laboratory that works as an online social network via blockchain-supported communication in Figure 13. In such a smart laboratory, the end-users, e.g., patients, can review clinical reports and connect with key stakeholders, e.g., insurance companies and doctors. Blockchain in clinical laboratory processes can be an interesting direction for the future. In the proposed model, the following applications can be considered in clinical laboratory systems:Clinical data sharing;Health consultation;Electronic health record (EHR) management;Pharmaceutical supply chain;Billing.

## 5. Conclusions

This paper presented an SLR on the Internet of Things in clinical laboratory systems. First, 2384 articles between 2015 and 2022 were screened based on the selection criteria. Next, 2121 studies were screened for titles and abstracts, of which 41 were screened for full-text review. A total of 18 articles were included in this SLR, and Cohen’s coefficient of the inter-reviewer agreement was 0.91. The included studies were divided into three classifications based on availability, including preanalytical, analytical, and postanalytical concerning RQ1. According to RQ2, common tools and environments were noted. A total of 78% of the included papers were real-tested. Communication-based approaches were the most common (83.3%), followed by application-based approaches (44.4%), and sensor-based approaches (33.3%) in the included studies. Both resource-based and security-based approaches were considered in 27.7% and 16.6% of the included investigations. According to RQ3, opportunities noted in the included studies were presented as a “smart lab” model. Finally, according to RQ4, open issues and challenges such as scalability, costs, energy consumption, interoperability, privacy and security, and performance should be considered. Certain limitations were revealed concerning the application of IoT-based laboratory tools that must be addressed in the future. We find in our review that there are multilayered challenges including costs, scalability, interoperability, security, and energy/power consumption, particularly in settings with inconsistent access to electricity, performance, and privacy. While these are certain limitations of the applications, future trends in clinical laboratory systems within the IoT ecosystem must focus on expanding the system’s scale (i.e., scalability) to ensure the adoption of the systems. Moreover, while the costs and energy consumption are high, this prominent open issue must be addressed in the future to ensure that the high operational costs are optimized for usage. The interoperability of different IoT systems ought to be addressed by ensuring that newly added IoT devices are compatible with the existing systems. Future studies must focus on the short-range to long-range networks including personal and cellular wireless communication networks to optimize power consumption with sensor devices. Additionally, cloud and grid computing channels can provide consistent access to data using connected devices with the option for decentralized storage to different users. Big data analytics is also an integrating tool that can enable huge volumes of data to be stored in a semistructured, structured, and unstructured format. When considering security issues, the IoT architecture ought to implement different layers (i.e., accuracy prevention and defense) to ensure the integrity of data systems. In summary, future trends include novel network integration, cloud and grid computing, fog computing, big data visualization and analytics, blockchain, and online social networks. With these results, we suggest that the IoT development of clinical laboratory systems be conducted more rigorously and efficiently.

## Figures and Tables

**Figure 1 sensors-22-08051-f001:**
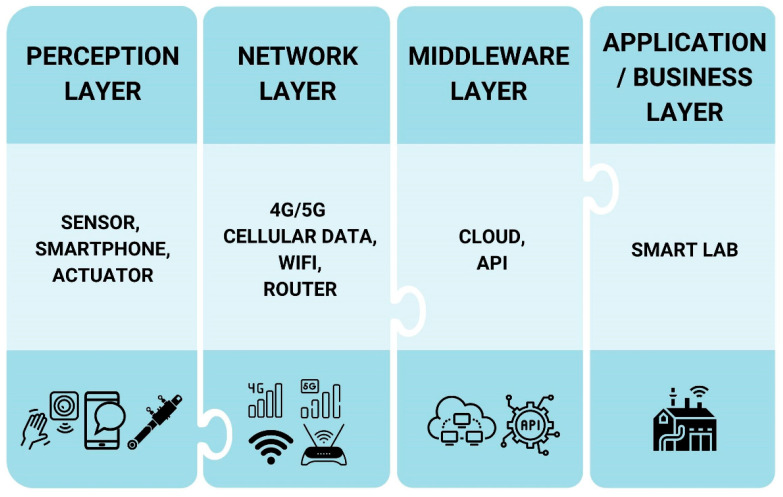
The architecture of IoT in clinical laboratories. Adapted from Ashton [31].

**Figure 2 sensors-22-08051-f002:**
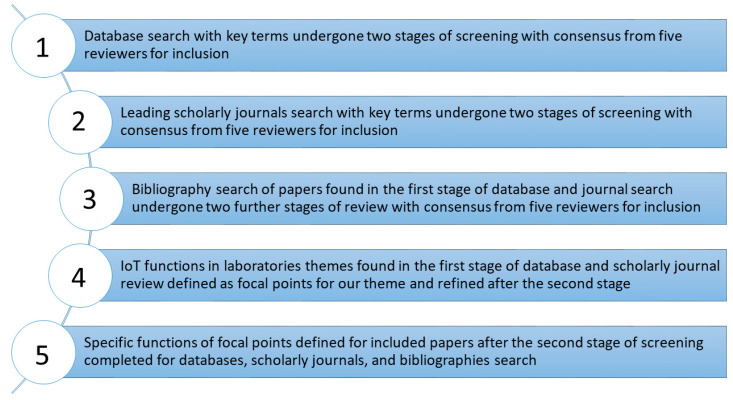
The concept-driven theoretical framework for the search strategy and selection of relevant investigations.

**Figure 3 sensors-22-08051-f003:**
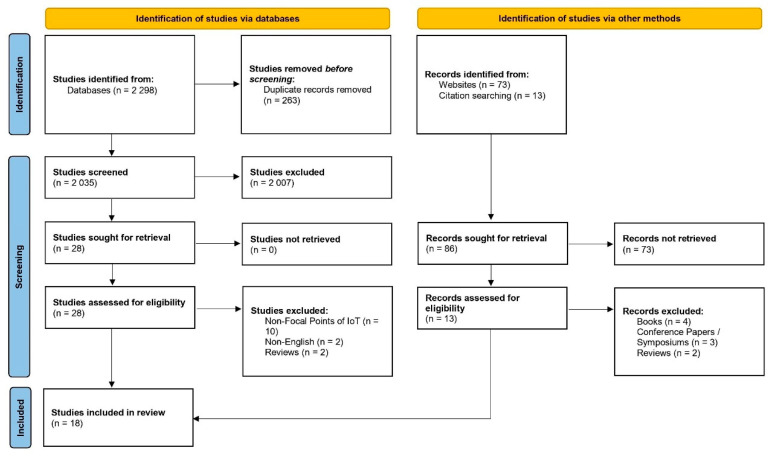
PRISMA flowchart.

**Figure 4 sensors-22-08051-f004:**
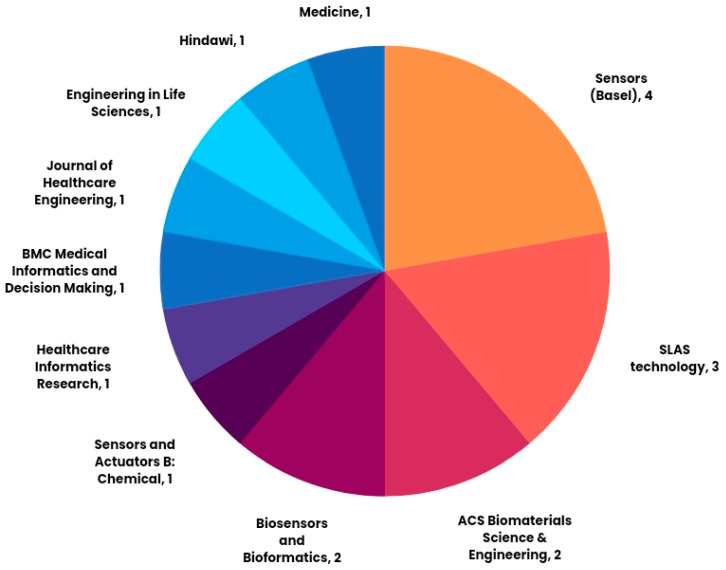
A pie chart of the number of investigations by journal name.

**Figure 5 sensors-22-08051-f005:**
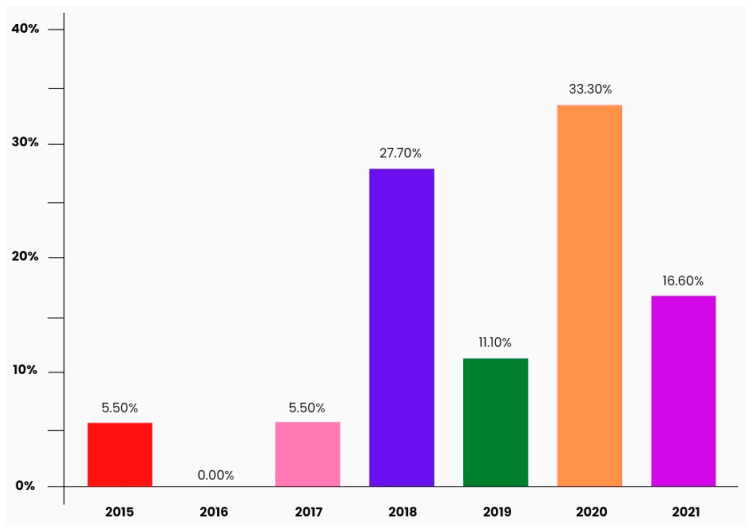
A bar chart of the percentage of investigations published each year since 2015.

**Figure 6 sensors-22-08051-f006:**
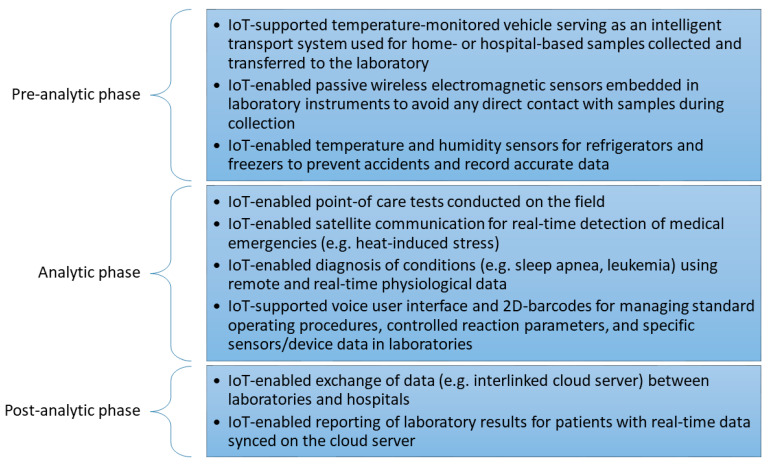
Specific functions of identified IoT focal points in all investigations’ clinical laboratory processes.

**Figure 7 sensors-22-08051-f007:**
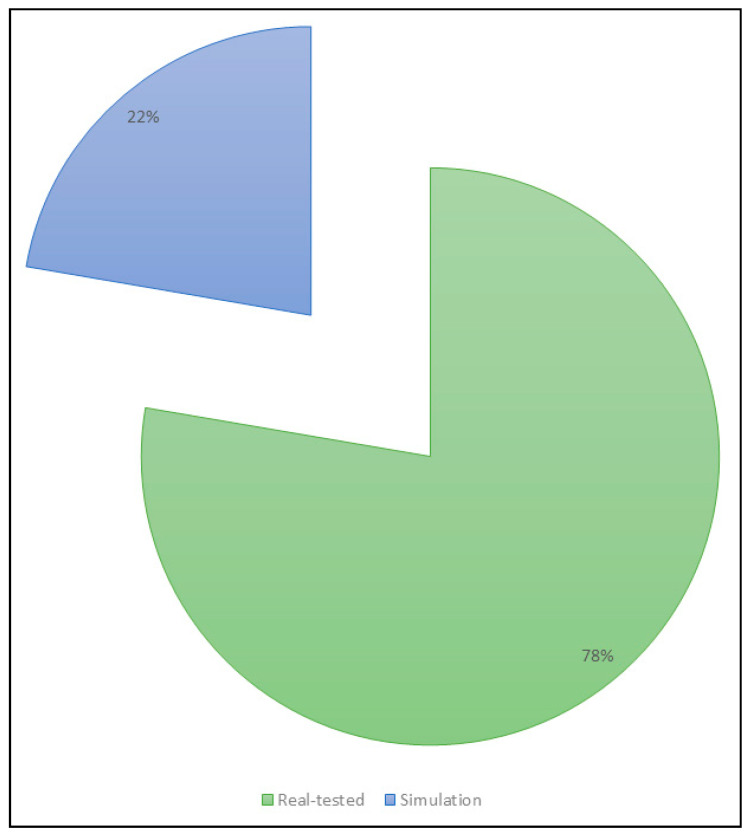
The percentage of evaluation techniques in all investigations.

**Figure 8 sensors-22-08051-f008:**
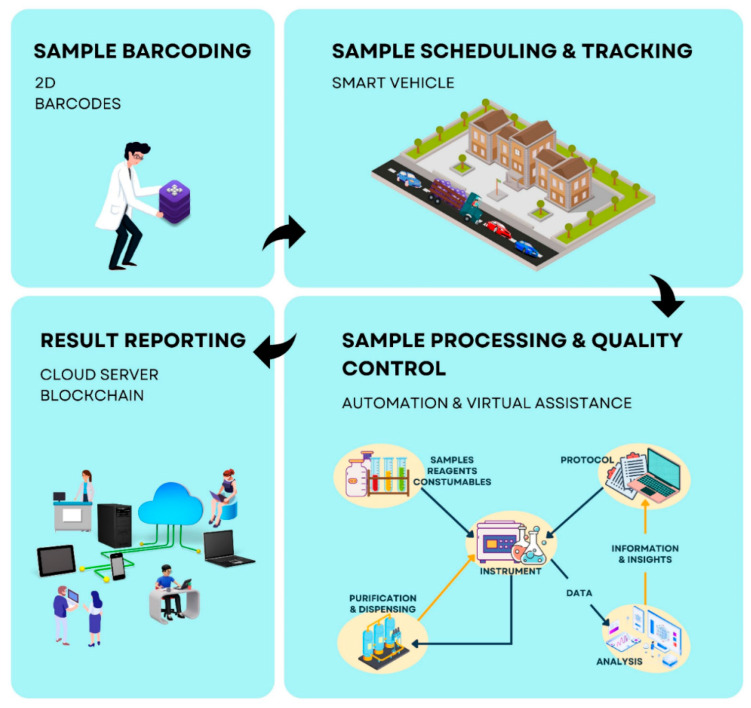
A depiction of a smart clinical laboratory supported by IoT.

**Figure 9 sensors-22-08051-f009:**
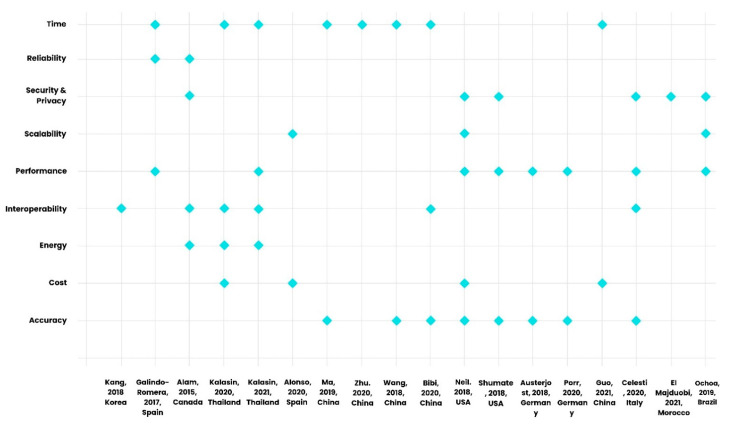
Advantages of IoT technologies from primary studies based on key metrics defined in Section 1.4, depicted in a scatter plot [56,57,58,59,60,61,62,63,64,65,66,67,68,69,70,71,72,73].

**Figure 10 sensors-22-08051-f010:**
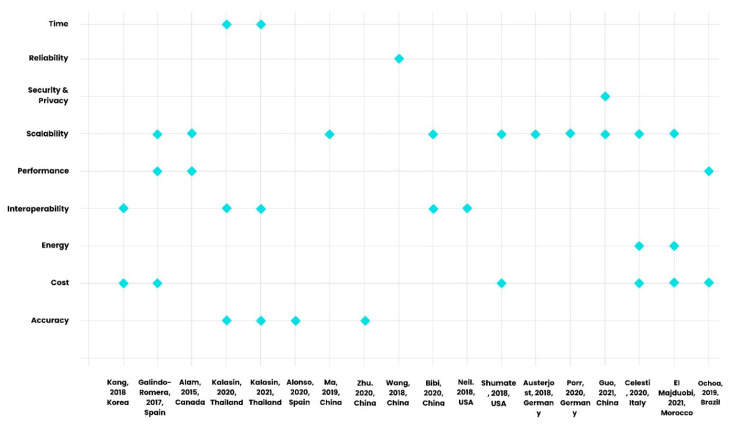
Challenges of IoT technologies from primary studies based on key metrics defined in Section 1.4, depicted in a scatter plot [56,57,58,59,60,61,62,63,64,65,66,67,68,69,70,71,72,73].

**Figure 11 sensors-22-08051-f011:**
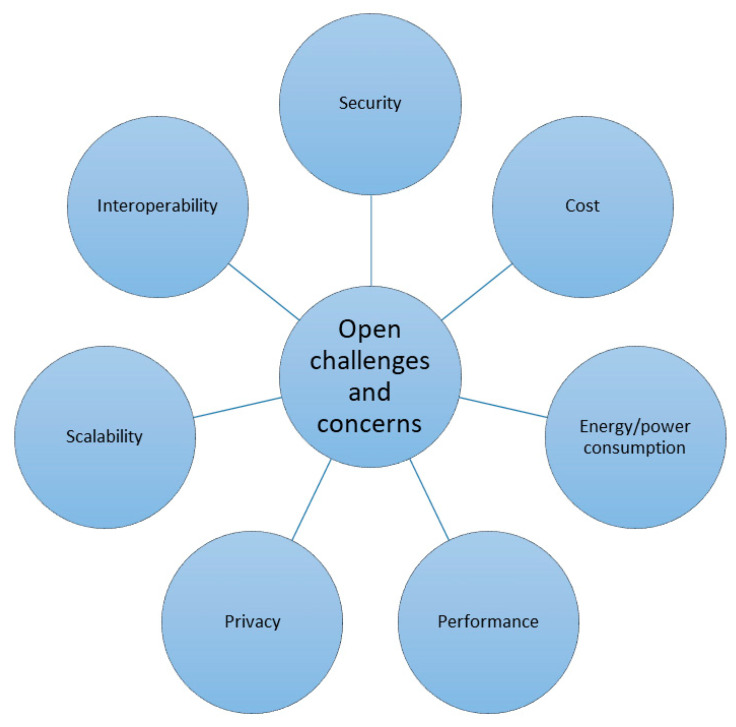
Open challenges and concerns are found in all investigations.

**Figure 12 sensors-22-08051-f012:**
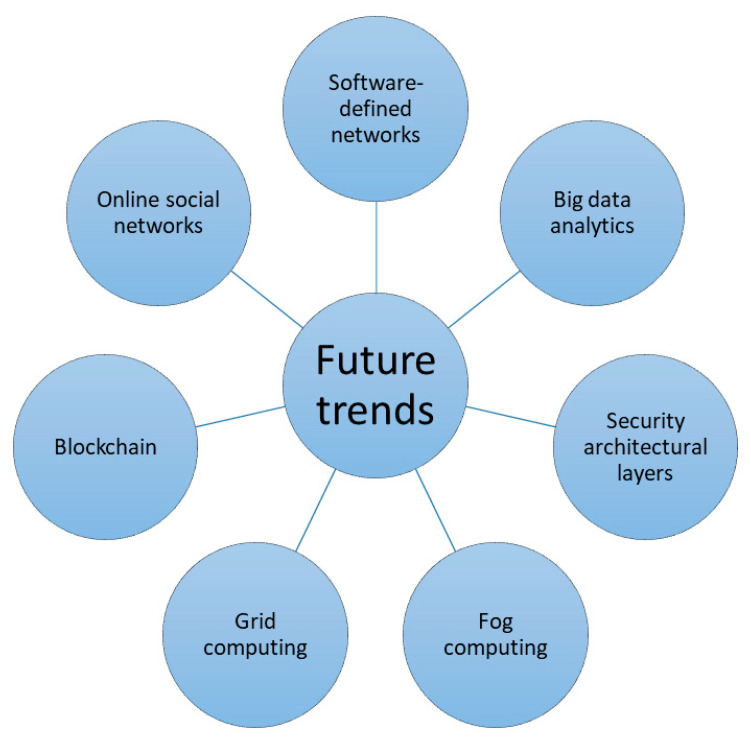
Future trends in clinical laboratory systems within IoT ecosystems.

**Figure 13 sensors-22-08051-f013:**
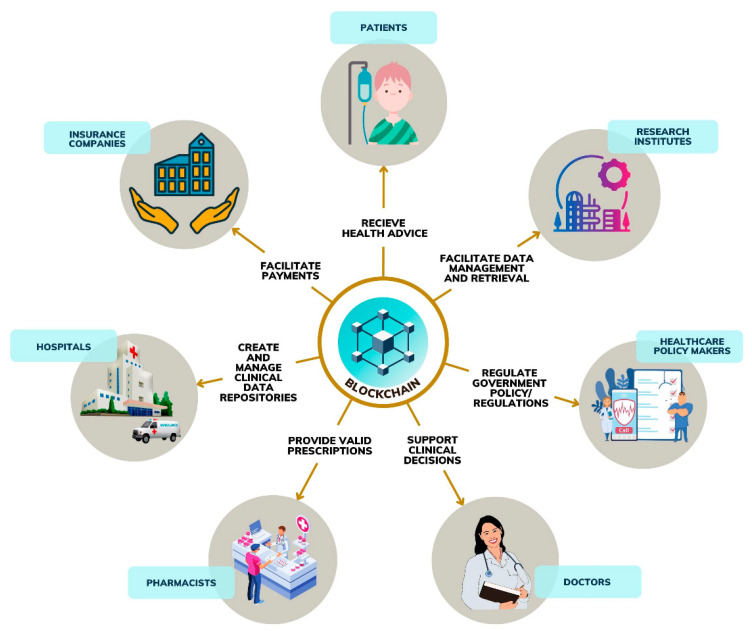
A blockchain-based approach to IoT-supported postanalytical dissemination of results in clinical laboratories.

**Table 1 sensors-22-08051-t001:** Literature review of related work.

Type of Study	Author, Year	Main Idea	Selection Strategy	Open Issue(s)	Taxonomy Provided
Survey	Thilakarathne, 2020, [42]	IoT in healthcare	Not specified	Discussed	No
	Dang, 2019, [43]	IoT and cloud computing for healthcare	Not specified	Discussed	No
	Islam, 2015, [44]	IoT in healthcare	Not specified	Discussed	Yes
	Dhanvijay, 2019, [45]	IoT and its applications in healthcare	Not specified	Discussed	No
Case study	Yuehong, 2016, [46]	IoT in healthcare	Not specified	Not discussed	No
	Habibzadeh, 2019, [47]	IoT in healthcare	Not specified	Discussed	No
Systematic literature review	Bolhasani, 2021, [48]	Deep learning with IoT in healthcare	Specified	Not discussed	Yes
	Forum, 2021, [49]	Blockchain integration in healthcare	Specified	Not discussed	No
	Sadoughi, 2020, [50]	IoT in medicine	Specified	Not discussed	No
	Dwivedi, 2022, [13]	IoT applications in smart healthcare	Specified	Discussed	No
	Nasiri, [51]	Security requirements of IoT-based healthcare system	Specified	Discussed	No

**Table 2 sensors-22-08051-t002:** Research questions and motivations.

Research Questions	Rationales
**RQ1**: What are the different categories and metrics for IoT integration in clinical laboratory systems?	The purpose is to gain insight into the scope and research scopes that support IoT-enabled clinical laboratory systems.
**RQ2**: What are the existing tools, evaluation techniques, and approaches that enable IoT in clinical laboratory systems?	The purpose is to identify and reveal existing tools and frameworks to advance the current status of IoT in clinical laboratory systems.
**RQ2**: What opportunities are available with the existing IoT techniques in clinical laboratory systems?	The purpose is to reveal the research opportunities in this field and present a coherent understanding of IoT applications in clinical laboratory systems.
**RQ3**: What are the key advantages, open challenges, concerns, and future trends in IoT in clinical laboratory systems?	The purpose is to identify the research considerations required for IoT applications in clinical laboratory systems.

**Table 3 sensors-22-08051-t003:** Summary of the framework/model, users, tools and evaluation environment, main idea, evaluation techniques, performance analysis, and opportunity of the included studies.

No	Primary Author, Year, Country	Journal	Framework/Model	User(s)	Evaluation Tool(s) and Environment(s)	Main Idea	Evaluation Technique(s)	Performance Analysis	Opportunity	Advantage(s)	Challenge(s)
**Preanalytic**											
1	Kang, 2018, Korea [56]	*Healthcare Informatics Research*	ZigBee communication	Scientists in the departments of laboratory, pathology, radiology, nuclear medicine, and radiation oncology	SMARTMIEW Cloud (Mbuzzer Co., Seoul, Korea) and Beagleboard (Texas Instruments, Dallas, TX, U.S.A.) using a temperature sensor	Able to maintain required temperature and humidity through a hospital IoT system implementation in the laboratory	Simulation, real-tested	Five issues were identified: sensing and measuring, transmission method, power supply, sensor module shape, and accessibility.	Equipment and environment monitoring in laboratories, device interconnection, and remote patient monitoring	High optimization	High maintenance costs, low reliability, high complexity
2	Galindo-Romera, 2017, Spain [57]	*Sensors* (Basel)	Portable Internet-of-Things (IoT) reader for passive wireless electromagnetic sensors based on a four-layer model	Lab scientists	Four functional layers: radiating layer, radiofrequency (R.F.) sensor interface, IoT mini-computer, power unit, MATLAB	Able to identify liquids without direct contact	Simulation, real-tested	A time difference between 3.59 ms and 8.04 ms for values of relative permittivity between 1 and 5.77, corresponding to free space and CHCl3	Measurement of different liquids without direct contact with readers, which may be used for lab spills with toxic substances	Low detection delay, high efficiency, high performance, low overhead time, low computation time	Low physical range, high costs, low scalability
3	Alam, 2015, Canada [58]	*Sensors* (Basel)	Cyber-physical cloud computing (CPCC) for social internet of vehicles (S.U.V.) model	Driver, passengers, social network portal user, transport authority, intelligent vehicle system	Android-based communication (android tablets, laptops, desktop computers), Java	Able to facilitate communication from vehicle-to-vehicle, vehicle-to-infrastructure, and vehicle-to-internet; store the information (e.g., safety, efficiency, and infotainment messages); and provide near-real-time data for vehicle users and offline use cases for smart behaviors of vehicles and transport authorities to promote intelligent transport systems	Simulation, real-tested	N.R.	Effective use of resources and multimedia sensory datasets allow for smart and scalable adaptation vehicles for home- or hospital-based laboratory samples by avoiding traffic jams or accidents	High optimization, high reliability, high efficiency, high security, high privacy	Low scalability, high computation overhead
**Analytic**											
4	Kalasin, 2020, Thailand [59]	*A.C.S. Biomaterials Science & Engineering*	Stress-free electrochemical-based sensor with IoT	Community health workers, community members	Smartphone, PSRAM-Wifi, Bluetooth, analytical-grade chemicals	Chemical transduction and Bluetooth-enabled P.O.C. technology	Simulation, real-tested	Sensitivity of 97.2% and data transmission was optimal	High outreach to underequipped communities of creatinine P.O.C. testing for those with kidney disorders with a minimal one-time investment	Low detection delay, high efficiency, low cost, high portability	Misuse, high maintenance, variable sensitivity
5	Kalasin, 2021, Thailand [60]	*A.C.S. Biomaterials Science & Engineering*	Satellite-based multimodal (artificial intelligence and IoT) sensing platform	Community health workers, community members	Smartphone, PSRAM-WIFI and Bluetooth, analytical-grade chemicals	Chemical transduction and Bluetooth-enabled P.O.C. technology	Simulation, real-tested	Selective efficacy of 96.3%	The concurrent monitoring of heat-stress sweat creatinine and the user’s heart rates allows for early diagnosis of heat stress through a decentralized epidermal sensor.	Low detection delay, moderate efficiency, low cost, high portability	Misuse, high maintenance, variable sensitivity
6	Alonso, 2020, Spain [61]	*Biosensors and Bioelectronics*	Field-programmable gate array (FPGA) with IoT capabilities	Community health workers, community members	Sensing system camera, IoT-enabled remote control, ELISA system	IoT-based malaria P.O.C. testing	Simulation, real-tested	Sensitivity, measured as photon detection probability, of the SPADs camera in a wavelength range of 300–1100 nm for overvoltages of 1.3 V and 1.8 V, at room temperature, was always below 15%	PoCT for plasmodium had similar signal trends and levels of detection to commercial fluorescence plate readers	Low cost, high scalability	Variable sensitivity
7	Ma, 2019, China [62]	*B.M.C. Medical Informatics and Decision Making*	Support vector machine-based strategy with IoT-enabled data monitoring	Healthcare personnel, community members	Smartphone, monitoring system, portable device, medical cloud monitoring center	Smartphone IoT-based real-time diagnosis of sleep apnea	Simulation	Sensitivity of 87.6%, accuracy of 90.2%, and specificity of 94.1%	Monitoring of sleep apnea in real-time with an early suggestion of abnormal physiological parameters	Low time, high sensitivity	Low scalability
8	Zhu, 2020, China [63]	*Sensors and Actuators B: Chemical*	IoT-enabled PCR chip	Community health workers, community members	Smartphone, PCR chip, D.N.A. templates/primers, Bluetooth, IoT cloud	Dengue virus P.O.C. testing	Simulation, real-tested	40 cycles to detect the cDNA of a DENV required 34 min	IoT-enabled PoCT for DENV can effectively detect complementary deoxyribonucleic acid (cDNA) with the potential to tackle infectious disease outbreaks	Low time	Variable sensitivity
9	Wang, 2018, China [64]	*Medicine*	Cloud-assisted IoT construction for strip test results processing	Healthcare personnel, community members	Cloud server, Strip test, IoT device	Lupus nephritis PoC testing with a cloud-assisted IoT	Simulation, real-tested	Sensitivity and specificity were both higher than 80%	A urinary biomarker with excellent potential for monitoring lupus nephritis was identified with IoT-enabled reporting	Low time, moderate sensitivity	Variable reliability
10	Bibi, 2020, China [65]	*Journal of Healthcare Engineering*	IoT-enabled microscope for blood smear image and leukemia subtype detection	Healthcare personnel	IoT-enabled microscope, datasets, data augmentation, leukemia cloud, dense convolutional neural network (DenseNet-121), and residual convolutional neural network (ResNet-34) models	IoT-based detection and classification of leukemia	Simulation	Accuracy [ResNet-34 Accuracy: ALL (100%), AML (99.65%), CLL (99.73%), CML (99.73%), Healthy (100%); DenseNet-121: ALL (100%), AML (99.91), CLL (99.91%), CML (100%), Healthy (100%)]	The suggested model is better than existent machine-learning algorithms for healthy-versus-leukemia-subtype identification	High accuracy, high interoperability, low time	Misuse, high complexity, low scalability
11	Neil, 2018, U.S.A. [66]	*SLAS Technology*	Lean process	Lab scientists	Biotage V10 centrifugal evaporator, BioMicroLab SampleScan 2D Plus, BlinkStick L.E.D. Strip, Raspberry Pi 3, iHome iM59 Rechargeable Mini Speaker, T.W.D. TradeWinds 30 mL EPA-Style Vial with Data Matrix 2D Barcodes, Biotage V10 Centrifugal Evaporator, Python, Java	Able to track purification analysis from beginning to end	Simulation, real-tested	N.R.	Tracking of laboratory samples in a high-throughput screening	Low costs, high efficiency, high scalability, high performance, high reliability	High communication complexity
12	Shumate, 2018, USA [67]	*SLAS Technology*	Computer-aided design model for 3D-printed components	Lab scientists	Arduino Uno, Ethernet Shield, RS232 Shield, Raspberry Pi, Java	Able to provide real-time gravimetric summaries of dispensing and generate timely alerts if problems arise	Simulation, real-tested	Average difference of 1.0% between the weights measured by the JAMS platform and the manual operator weighing	Monitoring the dispense rate of liquids in a high-throughput screening	High efficiency, high performance, high reliability	High costs, low scalability
13	Austerjost, 2018, Germany [68]	*SLAS Technology*	Voice user interface	Lab scientist	Node-RED 0.15.2, Java, Amazon’s Alexa Voice Service	Virtual assistant able to control and monitor smart devices in the laboratory	Simulation, real-tested	Mean accuracy of 95% ± 3.62 of speech command recognition	Instrument control, retrieval of experimental data, and protocols to improve workflows	High sensitivity, high performance	Low scalability
14	Porr, 2020, Germany [69]	*Engineering in Life Sciences*	Open source Standardization in Lab Automation 2 standard	Lab scientist	Wireless Local Area Network, ESTful Hypertext Transfer Protocol, ControlFlow Runtime, REST-API	The central lab server facilitates device communication and database records of measurements, tasks, and results generated.	Simulation	N.R.	Digitalization of workflow in laboratories enabling device communication and automation	High efficiency, high performance	Low scalability
**Postanalytical**											
15	Guo, 2021, China [70]	*Biosensors and Bioelectronics*	5G-enabled fluorescence sensor with cloud infrastructure as a service (IaaS)	Patients, healthcare personnel, medical facilities, government	Hardware devices (personal computers, 5G smartphones, IPTV), 5G cloud server	Quantitative detection of spike protein and nucleocapsid protein of SARS-CoV-2 by using mesoporous silica encapsulated up-conversion nanoparticles (UCNPs@mSiO2) labeled lateral flow immunoassay (LFIA), of which medical data can be transmitted to the fog layer of the network and 5G cloud server for edge computing and big data analysis	Simulation	The sensor can detect spike protein (S.P.) with a detection of limit (L.O.D.) of 1.6 ng/mL and nucleocapsid protein (N.P.) with an L.O.D. of 2.2 ng/mL	A practical and efficient way to treat and prevent COVID-19 and other mass infectious diseases in the future	High optimization, low latency, high reliability, low communication cost	Low scalability, low privacy
16	Celesti, 2020, Italy [71]	*Sensors* (Basel)	Blockchain technology with cloud infrastructure as a service (IaaS)	Patients, healthcare personnel, medical facilities, government	Python, Ethereum, and blockchain	Tele-medical laboratory service where technicians perform clinical exams on patients directly in a hospital through IoT medical devices and results are automatically sent via the hospital cloud to doctors of federated hospitals for validation or consultation.	Simulation, real-tested	The average cost of a single transaction to be written in the public Ethereum blockchain is approximately 0.0002 ETH	Virtual telemedical laboratory service may be created through a healthcare workflow operating in a federal hospital IoT cloud environment leveraging blockchain	High transparency, high security, high privacy, high traceability, high efficiency	High cost, low scalability, high energy utility
17	El Majdoubi, 2021, Morocco [72]	*Hindawi*	Data sharing through an end-to-end blockchain-based and privacy-preserving framework called SmartMedChain	Patients, healthcare personnel, medical facilities, government	Blockchain	Blockchain-based framework with encrypted health data stored in interplanetary file system (IPFS) that supports data access and usage for smart healthcare	Simulation, real-tested	Average throughput of 39.6 tps and an average delay of 1.34 s at 50 tps workload	Laboratory data integration for improved healthcare provision	High transparency, high security, high privacy, high traceability, high efficiency	High cost, low scalability, high energy utility
18	Ochôa, 2019, Brazil [73]	*Sensors* (Basel)	UbiPri middleware	Patients, healthcare personnel, medical facilities, government	Ethereum blockchain, IPFS storage service, middleware, 16-bit Arduino Uno R3 architecture	A smart contract between different architectural layers of IoT, including Ethereum Blockchain Smart, for user privacy	Simulation, real-tested	The user transaction fee was 0.06779326 ETH and 3,389,663 gas was used; the device transaction fee was 0.0277857 ETH and 1,389,285 gas was used; environment transaction fee was 0.02598354 and 1,299,177 gas was used.	Privacy for blockchain architectures in IoT health environments	High scalability, high performance, high privacy	High cost, high energy utility

**Table 4 sensors-22-08051-t004:** Summary of the approaches and scopes of the included studies, based on key approaches summarized in Section 1.3.

No.	Primary Author, Year, Country	Approach	Scope
1	Kang, 2018, Korea [56]	Sensor-based	Environmental sensors
		Communication-based	Algorithmic approach
2	Galindo-Romera, 2017, Spain [57]	Sensor-based	Environmental sensors
		Communication-based	Technological approach
3	Alam, 2015, Canada [58]	Resource-based	Scheduling
		Sensor-based	Environmental sensors
		Communication-based	Algorithmic approach
4	Kalasin, 2020, Thailand [59]	Application-based	Prediction/diagnosis
		Sensor-based	Wearable sensors
5	Kalasin, 2021, Thailand [60]	Application-based	Prediction/diagnosis
		Sensor-based	Wearable sensors
		Communication-based	Technological approach
6	Alonso, 2020, Spain [61]	Application-based	Prediction/diagnosis
		Communication-based	Technological approaches
7	Ma, 2019, China [62]	Application-based	Prediction/diagnosis
		Communication-based	Technological approach
8	Zhu, 2020, China [63]	Application-based	Prediction/diagnosis
		Communication-based	Technological approach
9	Wang, 2018, China [64]	Application-based	Prediction/diagnosis
		Communication-based	Technological approach
10	Bibi, 2020, China [65]	Application-based	Algorithmic approach
		Communication-based	Technological approach
11	Neil, 2018, USA [66]	Sensor-based	Environmental approach
		Communication-based	Technological approach
		Resource-based	Resource allocation, offloading, load balancing, and provision
12	Shumate, 2018, USA [67]	Sensor-based	Environmental approach
		Communication-based	Algorithmic approach
		Resource-based	Resource allocation, offloading, load balancing, and provision
13	Austerjost, 2018, Germany [68]	Communication-based	Algorithmic approach
14	Porr, 2020, Germany [69]	Communication-based	Algorithmic approach
15	Guo, 2021, China [70]	Application-based	Prediction/detection
16	Celesti, 2020, Italy [71]	Security-based	Access control
		Resource-based	Scheduling
		Communication-based	Algorithmic approach
17	El Majdoubi, 2021, Morocco [72]	Security-based	Privacy
		Resource-based	Scheduling
		Communication-based	Algorithmic approach
18	Ochôa, 2019, Brazil [73]	Security-based	Privacy

## Data Availability

All data utilized in this study are available online.

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
