# Peer review of "A Systematic Review of Internet of Things in Clinical Laboratories: Opportunities, Advantages, and Challenges"

_sensors, 2022, doi:10.3390/s22208051_

Round 1

Reviewer 1 Report

The last two years have proved that technology can be crucial in the field of healthcare. In particular, emerging proposals based on the Internet of Things (IoT) offer a wide variety of applications. Authors are aware of this fact and have compiled a huge number of references, covering different applications and proposals of IoT technologies for healthcare.

The paper is a very extensive and complete survey of up-to-date articles about the topic. More than 2300 articles have been reviewed and taken into account, according to authors, and classified with respect to the application field.

Results of the study are presented in a large table. The table is difficult to read. It is true that it must summarize a large amount of information, but it should be presented in a more comfortable way, if possible.

Figure 5 in page 27 should be presented using different colors, not only variations of blue, in order to make it easier to visualize.

Throughout the text some technologies are mentioned several times: Cloud computing, Fog computing, Big Data, PoC testing... Sometimes, the terms are capitalized and other are not. In the same sense, some paragraphs starts with an indent, while other do not. The document should follow an homogeneous format.

Author Response

Reviewer 1 Comments and Author Responses:

The last two years have proved that technology can be crucial in the field of healthcare. In particular, emerging proposals based on the Internet of Things (IoT) offer a wide variety of applications. Authors are aware of this fact and have compiled a huge number of references, covering different applications and proposals of IoT technologies for healthcare.

The paper is a very extensive and complete survey of up-to-date articles about the topic. More than 2300 articles have been reviewed and taken into account, according to authors, and classified with respect to the application field.

Reviewer 1, Comment 1: Results of the study are presented in a large table. The table is difficult to read. It is true that it must summarize a large amount of information, but it should be presented in a more comfortable way, if possible.

Author Response: In line with your comments, Tables 5-6 have been converted to Figures and more supporting text has been added. The tables have been summarized to make them more comfortable. Beyond this, it would not be possible to shorten them further.

Reviewer 1, Comment 2: Figure 5 in page 27 should be presented using different colors, not only variations of blue, in order to make it easier to visualize.

Author Response: The figure has been updated with new colors and it has been made clearer for better visualization.

Reviewer 1, Comment 3 Throughout the text some technologies are mentioned several times: Cloud computing, Fog computing, Big Data, PoC testing... Sometimes, the terms are capitalized and other are not. In the same sense, some paragraphs starts with an indent, while other do not. The document should follow an homogeneous format.

Author Response: All discrepancies have been fixed. Please have a look. 

Reviewer 2 Report

The paper conducted a systematic literature review related to the Internet of Things in clinical 2 laboratories which is very interesting. However, the paper presentation and organization can be improved. I have the following suggestions:

1. Introduction needs to be revised to include a brief background about the title, the main problems that motivates the author to conduct this study, the main objectives of the study, a brief description of the method to achieve the objectives, and the organization of the paper.
2. What is the purpose of Figure 2? It just shows some keywords that can be added in the text as the point forms. If you want to keep this figure, you need to add more details.
3. I suggest to add a section as background or overview and move section 1.1 to 1.5 under this background section.
4. Motivation of the study needs to be based on the shortcomings of the existing studies.
5. Do you need to have Figure 3 since the main important Figure for the systematic literature review is Figure 4 PRISMA flowchart.
6. Please add the list of keywords and searching strategies.
7. Improve the quality of the figures. Some of them are blur such as Figure 5 and 6.
8. The answer for RQ1 is too short. Please add more discussion.
9. Add more discussion for Table 5 and 6. You can replace them with bar charts or any other figures as well.
10. Conclusion needs to include the main contributions of the paper, the concluding remarks that you have made after finalizing this study, the limitations and future direction. Concluding remarks and limitations are missing from the current conclusion section.

Author Response

Reviewer 2 and Author Responses:

The paper conducted a systematic literature review related to the Internet of Things in clinical 2 laboratories which is very interesting. However, the paper presentation and organization can be improved. I have the following suggestions:

Reviewer 1, Comment 1: Introduction needs to be revised to include a brief background about the title, the main problems that motivates the author to conduct this study, the main objectives of the study, a brief description of the method to achieve the objectives, and the organization of the paper.

Author Response: These sections are already present, but some changes are being made. Please ensure to review the paper in full for the introduction. We have already specified the motivation, objectives and method to achieve this objective.

Motivation: The motivation to conduct an S.L.R. on IoT in clinical laboratories is that until now, there has been no systematic review of IoT-based clinical laboratories. Most existing papers focus on healthcare systems holistically, and these studies do not analyze trends, open issues, and challenges of IoT in clinical laboratories. Many of the studies summarized in section 1.4 do not provide open challenges. Ours is the first paper to conduct an S.L.R. to explore IoT in clinical laboratories. Therefore, we conducted a comprehensive review to answer the research questions and associated motivations in the present study (Table 2).

Objective: A comprehensive systematic review of IoT in clinical laboratories has not been con-ducted. With the implementation of IoT in healthcare already in motion, a robust understanding of the clinical laboratories' IoT adaptation is necessary. We believe that a comprehensive review inclusive of IoT priority areas in laboratories will provide an integrated perspective and serve as a repository for knowledge. In the present study, we aimed to collate all existing evidence, classifications, and applications of IoT in dif-ferent clinical laboratory processes. Our objectives were to identify, classify, and evaluate the current IoT literature to inform our understanding of the IoT-supported clinical laboratory adoption and implementation.

Method to achieve the objectives: This structured, systematic review aims to identify, classify, and evaluate IoT applications in clinical laboratory systems. The first section describes the introduction, defini-tion of IoT in section 1.1, IoT architecture in section 1.2, approaches and scopes of IoT in section 1.3, IoT metrics in section 1.4, review of related literature in section 1.5, aim and objective in section 1.6, and motivation for this study in section 1.7. The second section details the methods, including the theoretical framework and search strategy in section 2.1, eligibility criteria in section 2.2, data extraction and synthesis in section 2.3, and critical appraisal and bias assessment in section 2.4. The third section describes the results, including the risk of bias in section 3.1, IoT in the preclinical laboratory phase in section 3.2, IoT in the clinical laboratory phase in section 3.3, and IoT in the post-analytical laboratory phase in section 3.4. The fourth section details the discussion, including open concerns, challenges, and future trends in section 4.1. The conclusion is summarized in section 5.

Reviewer 2, Comment 2: What is the purpose of Figure 2? It just shows some keywords that can be added in the text as the point forms. If you want to keep this figure, you need to add more details.

Author Response: The figure has been removed entirely.

Reviewer 2, Comment 3. I suggest to add a section as background or overview and move section 1.1 to 1.5 under this background section.

Author Response: Your comment is unclear. Based on what you have written, you are suggesting that we move section 1.1-1.5 under a background section? This already falls under the background/introduction.

Reviewer 2, Comment 4. Motivation of the study needs to be based on the shortcomings of the existing studies.

Author Response: More information has been added on shortcomings of the existing studies under the section “motivation:”

As highlighted in the assessment of current literature, existing surveys, case studies and systematic literature reviews that are conducted for IoT in healthcare do not depict robustness in approaching IoT in the clinical laboratories’ taxonomy. Hence, first-ly, this study aims to robustly identify the different IoT categories and metrics to integrate them in clinical laboratory systems. Secondly, this study also wishes to identify and reveal the tools and frameworks that can advance the current status of IoT in clinical laboratory systems. Thirdly, this study will also reveal the opportunities present in this field to have a coherent understanding of IoT applications. Lastly, this pa-per will identify the research considerations required to apply IoT in clinical laborites.

Reviewer 2, Comment 5. Do you need to have Figure 3 since the main important Figure for the systematic literature review is Figure 4 PRISMA flowchart.

Author Response: Yes, Figure 3 is the backend process of the methods, whereas the PRISMA flow chart only depicts the study selection process. Both are essential and we believe our study requires both figures.

Reviewer 2, Comment 6. Please add the list of keywords and searching strategies.

Author Response: They are attached in Supplementary Table 1 and have been annotated in the paper as well.

Reviewer 2, Comment 7. Improve the quality of the figures. Some of them are blur such as Figure 5 and 6.

Author response: the figures have been updated to more clear variants.

Reviewer 2, Comment 8. The answer for RQ1 is too short. Please add more discussion.

Author Response: More discussion has been added. Please refer to the text highlighted in yellow.

Reviewer 2, Comment 9. Add more discussion for Table 5 and 6. You can replace them with bar charts or any other figures as well.

Author Response: Thank you for the suggestion. Tables 5-6 have been converted to scatter plots and more text has been added to discuss the findings.

Reviewer 2, Comment 10. Conclusion needs to include the main contributions of the paper, the concluding remarks that you have made after finalizing this study, the limitations and future direction. Concluding remarks and limitations are missing from the current conclusion section.

Author Response: More clarification for the stated information has been added to the conclusion. Highlighted in yellow.